# Structural Studies of Bacteriophage Φ6 and Its Transformations during Its Life Cycle

**DOI:** 10.3390/v15122404

**Published:** 2023-12-11

**Authors:** J. Bernard Heymann

**Affiliations:** 1National Institute of Arthritis and Musculoskeletal and Skin Diseases, National Institutes of Health, 50 South Dr., Bethesda, MD 20892, USA; heymannb@mail.nih.gov; Tel.: +1-301-846-6924; 2National Cryo-EM Program, Cancer Research Technology Program, Frederick National Laboratory for Cancer Research, Leidos Biomedical Research, Inc., Frederick, MD 21701, USA

**Keywords:** *Cystoviridae*, cryoEM, dsRNA, ssRNA, RNA-dependent RNA polymerase, virus capsid, virus envelope, virus infection

## Abstract

From the first isolation of the cystovirus bacteriophage Φ6 from *Pseudomonas syringae* 50 years ago, we have progressed to a better understanding of the structure and transformations of many parts of the virion. The three-layered virion, encapsulating the tripartite double-stranded RNA (dsRNA) genome, breaches the cell envelope upon infection, generates its own transcripts, and coopts the bacterial machinery to produce its proteins. The generation of a new virion starts with a procapsid with a contracted shape, followed by the packaging of single-stranded RNA segments with concurrent expansion of the capsid, and finally replication to reconstitute the dsRNA genome. The outer two layers are then added, and the fully formed virion released by cell lysis. Most of the procapsid structure, composed of the proteins P1, P2, P4, and P7 is now known, as well as its transformations to the mature, packaged nucleocapsid. The outer two layers are less well-studied. One additional study investigated the binding of the host protein YajQ to the infecting nucleocapsid, where it enhances the transcription of the large RNA segment that codes for the capsid proteins. Finally, I relate the structural aspects of bacteriophage Φ6 to those of other dsRNA viruses, noting the similarities and differences.

## 1. Significance of Bacteriophage Φ6

The members of the *Cystoviridae* are the only known double-stranded RNA (dsRNA) bacteriophages, with bacteriophage Φ6 being the first isolated and the type species of the family [1]. Additional members of the family are being added [2], including bacteriophages Φ8, Φ12, and Φ13, as mentioned in this review. They represent counterparts to the dsRNA viruses, infecting eucaryotes and offer simpler systems for studies of structure and life cycle. These involve questions such as how the genome is encapsidated in the protective virus shells [3] and the roles of the component proteins in the propagation of the virus. Such insights may lead to the development of antivirals and treatments for managing pathogenic viruses [4]. In addition, the presence of dsRNA is typically a sign of viral infection [5].

The host of bacteriophage Φ6, *Pseudomonas syringae* (previously known as *P. phaseolicola*), is a plant pathogen of commercial importance in agriculture. One potential use for such a phage is therefore in controlling bacterial infections in plants [6,7].

In recent years, bacteriophage Φ6 has been used as a surrogate for enveloped viruses such as SARS-CoV-2 [4]. Used with care to acknowledge differences [8], it can be a valuable tool to understand environmental factors associated with viral infectivity without undue exposure to a dangerous virus.

## 2. Life Cycle of Bacteriophage Φ6

The Φ6 virion is composed of multiple layers, with the P1 protein forming an icosahedral capsid shell enclosing the dsRNA genome, in turn surrounded by an intermediate shell composed of proteins P4 and P8, and finally covered by an outer protein-lipid shell or envelope (Figure 1). The genome itself is divided into three parts: small (S), medium (M), and large (L) segments of dsRNA (the ssRNA transcripts are denoted as s, m, and l in lowercase. See [9]. for examples). The segments code for proteins at different stages of the life cycle (Figure 2) [10]. Upon infection, virions first bind to the host pili [1], requiring the protein P3 anchored onto P6 [11]. The pili then retract, and the virions fuse with the outer membrane mediated by the P6 protein [11]. The lysin (protein P5) degrades the peptidoglycan layer [12,13], allowing viral particle access to the inner membrane, which can then be traversed in an energy-dependent manner [14]. The P8 layer is subsequently stripped off and degraded [14]. What remains is the nucleocapsid, decorated with the P4 protein on the outside, producing ssRNA transcripts. A host protein, YajQ, binds to the outside of the nucleocapsid to enhance the transcription of the L segment [15]. The transcripts from all three segments are exported and translated into different viral proteins by the host machinery.

The four translation products of the l segment, P1, P2, P4, and P7, assemble into procapsids [18,19]. The three ssRNA segments are then sequentially packaged into the capsid [20], with concurrent expansion. Minus strand synthesis (replication) follows the completion of the packaging of the l segment [9]. The P8 layer is then assembled onto the new nucleocapsids [14]. The non-structural protein, P12, is required by P9 to form the proteolipid envelope, incorporating proteins P3, P6, P10, and P13 [21]. The proteins P5 and P10 cause cell lysis and the release of progeny virions [12,22].

The quantitation of the copy number of every protein is important for understanding the assembly and transformations of the virion and its sub-particles during the life cycle [23]. In the following section, I will cover not only the structure but also the copy numbers of the various proteins and how that relates to function. There will also be references to the structures of related viruses, such as Φ8 and Φ12. These show some sequence similarity but with very similar genomic organization protein structures [2].

## 3. The Initial Assembly Product, Procapsid

The procapsid (also called the polymerase complex) consists of four proteins [18]: P1, which makes up the icosahedral shell; P2, the RNA-dependent RNA polymerase (RdRP) [24]; P4, the packaging motor [25,26]; and P7, an assembly and packaging enhancer [27,28]. Early attempts to crystallize the procapsid failed, opening the way for studies by electron microscopy. As will be discussed later, the procapsid is in a metastable state, and the P4 protein is a hexamer in a symmetry-mismatch position weakly bound to the P1 shell. These features may not be conducive to stable crystal contacts and required imaging by electron microscopy of fresh samples soon after preparation.

### 3.1. The P1 Shell

The main part of the procapsid consists of an icosahedral shell formed by the 85 kDa P1 protein [29]. Early cryoEM reconstructions revealed a compact particle with invaginated five-fold vertices, giving it a star-like appearance (Figure 3A) [30,31,32,33,34,35].

The stoichiometry of the virion indicated that there are 120 copies of the P1 protein per particle [23], which means there are two copies per asymmetric unit, similar to other dsRNA viruses (Figure 3C) [30,36]. Better cryoEM reconstructions allowed for the discernment of structural details of the two copies [31] and eventually to near-atomic detail [37]. The architecture is very similar in the related bacteriophage Φ8 [16].

While the whole procapsid could not be crystallized, a pentamer of P1 in the presence of P7 was solved (Figure 3B) [37]. Similarly, a pentamer of the P1 from Φ8 was also solved by X-ray crystallography [38]. The pentamer forms the invaginated five-fold vertices, and the subunits are designated as P1_A_ (blue in Figure 3C). The other subunits are designated as P1_B_, connecting the two-fold and three-fold axes (red/orange/yellow in Figure 3C). P1_A_ and P1_B_ have identical folds with only minor differences in conformation to adapt to their different positions in the capsid (an example of “quasi-equivalence” of the subunits of the capsid [39]) [37].

**Figure 3 viruses-15-02404-f003:**
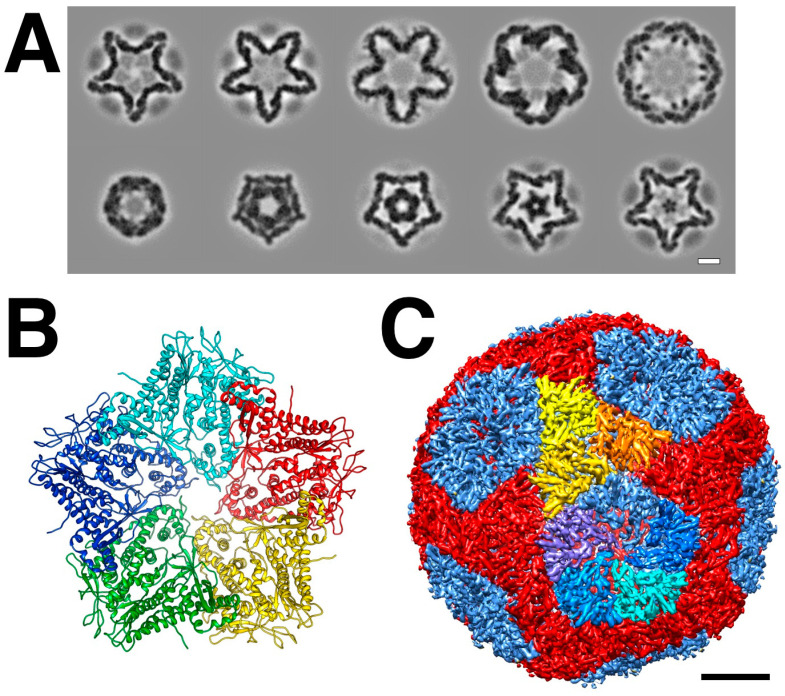
The procapsid shell. (**A**) Sections through a procapsid showing the invaginated five-fold vertices of the P1 shell. The diffuse densities are the minor procapsid proteins. (EMDB: 1501) [32] (**B**) The pentamer of P1 solved by X-ray crystallography. (PDB: 4K7H) [37] (**C**) Distinction of the P1_A_ (blues) and P1_B_ (red, orange, yellow) subunits in the procapsid. Scale bars: 100 Å.

### 3.2. The Packaging Motor, Protein P4

The 35 kDa protein P4 was identified as an NTPase required for the packaging of positive ssRNA strands into the procapsid [25,26]. The P4 protein forms a hexamer [40,41] that sits in the five-fold invagination in the procapsid with an obvious symmetry mismatch [42]. Both in virions, as well as recombinant and assembled procapsids, about 60–70 P4 monomers per particle were reported, close to the expected 72 (12 hexamers) [43]. In bacteriophage Φ8 nucleocapsids, the average occupancy of P4 on the five-fold vertices were found to be ~60 [44]. However, isolated Φ6 procapsids in other studies consistently had lower copy numbers [33,34,35]. Increased ionic strength, heat, and acidic conditions lead to P4 dissociation and procapsid expansion [33,45,46]. The interaction between P4 and P1 is therefore rather weak, perhaps tuned to intracellular conditions differing significantly from those in isolates.

Both available high-resolution procapsid reconstructions have very low P4 density [37,47], precluding assessing its association with the P1 shell. It is likely that the same interaction is maintained in the nucleocapsid, where we have several reconstructions. An asymmetric reconstruction of the five-fold vertex shows the P4 hexamer on top of the P1 shell and surrounded by the P8 layer (Figure 4A,C) [48]. Because of the symmetry mismatch, only some of the protrusions from the hexamer are expected to interact with P1. In Figure 4B, the thresholding of the isosurface was adjusted to show four possible connections. Sun et al. [48] identified a density not attributed to the P1 shell but consistent with the C-terminal tail of P4 (Figure 4D). The C-terminal 13 residues were previously shown to be important for procapsid assembly [49]. This tail extends over both the P1_A_ and P1_B_ subunits. The density is strong enough to suggest that five of the six P4 C-termini bind to the P4 shell. In procapsids, P4 is often shifted slightly from the five-fold axis [33], suggesting that some of its C-terminal tethers may be less tightly bound.

### 3.3. The RNA-Dependent RNA Polymerase, Protein P2

The whole Φ6 procapsid was functionally referred to as the polymerase complex because of its activity in transcription and replication [50,51]. The 75 kDa protein P2 was identified as the actual polymerase responsible for both replication and transcription [24], and its structure was solved by X-ray crystallography [52].

The polymerase is self-priming, with an emphasis on the first two nucleotides at the 3′ of the ssRNA template, fitting into a pocket of the C-terminal domain [52,53]. It can replicate a variety of ssRNA and is stimulated by manganese (II) (Mn^2+^) [24]. Mn^2+^ appears to lessen the specificity of the polymerase for the 3′ terminal sequence [54] by decreasing stability [55], suggesting a mechanism for regulation of its activity. Mn^2+^ consistently enhances activity of polymerases from the related bacteriophages Φ6, Φ8, and Φ13, but with different magnitudes [56], reflecting subtle variations in structure. Self-priming is dependent on the presence of the C-terminal domain linked through residues 601–614 [53]. Mn^2+^ destabilizes the interface of the C-terminal domain to allow for permissive elongation [57]. An amino acid substitution in the Mn^2+^ binding site (E491Q) stabilizes the C-terminal domain, desensitizing it to the stimulatory effect of Mn^2+^ [55]. This becomes more important in the context of the infection cycle, where Φ6 developed a mechanism for controlling the production of the l transcript by exploiting the host protein, YajQ (see Section 6) [58].

In the early cryoEM reconstructions of the procapsid, P1 and P4 were readily assigned to densities [42], but the locations of both P2 and P7 were not evident. The P2 protein was only located when it was noticed that there is a persistent density on the 3-fold axis inside the procapsid (Figure 5A–C) [32]. Because it exists as a monomer, there is a symmetry mismatch and its density in the cryoEM map is reduced, explaining why it was originally missed. An asymmetric cryoEM reconstruction of the location of P2 on the three-fold axis yielded a map that provided the proper orientation of the molecule (Figure 5D,E) [47]. Specific interactions with P1 in the palm and finger domains were confirmed by mutational analysis [59].

Knowing where P2 is located, it can be identified in cryo-electron tomograms of individual procapsids, allowing quantitation by counting [33]. The average occupancy was found to be about 8 out of the 20 possible sites with a distribution that indicates random incorporation during procapsid assembly. This agrees with dose-dependent incorporation during in vitro assembly of procapsids [43,59].

### 3.4. The Packaging Enhancer, Protein P7

The 17 kDa protein P7 enhances assembly of the procapsid and packaging but is not essential in either function [27,28,43,60]. The location of P7 in the procapsid was shrouded in mystery for a time. The only known structure is the N-terminal part of the homologous protein in bacteriophage Φ12 (Figure 6B) [61]. Because it was solved as a dimer, the initial assumption was that it would be a dimer in the procapsid. It was reported as such a dimer within the intermediate layer, outside the P1 shell [62]. Neutron scattering could only provide a rough radius for both P2 and P7, with P7 possibly located either inside or outside the capsid [63]. Earlier reconstructions of procapsids with and without P7 did not produce any meaningful results [32,42]. With better data and higher resolution maps, P7 was eventually located inside P147 procapsids on the sides of the invaginated five-fold vertices (Figure 6A) [34,35]. The density suggests an orientation of P7 based on the partial structure from Φ12 (Figure 6C).

These sites overlap with the location of P2, indicating that the substoichiometric amounts are the result of competitive binding [35]. In assembled procapsids with a large excess of P7, ~60 copies of P7 incorporate into each particle, and the inclusion of P2 is suppressed [43]. However, at about a stoichiometric ratio, the copy number was only ~45 for P7 and 11 for P2 [43], closer to the numbers for recombinant procapsids. From these findings, the expected maximum copy numbers are ~30 for P7 and ~10 for P2, with any excess attributed to non-binding inclusion in in vitro assembled particles. Both these proteins appear to be functional in substoichiometric amounts.

### 3.5. Procapsid Assembly

As is common for many other virus capsids, the procapsids of Φ6 readily assemble from recombinant constructs in vivo or purified proteins in vitro. The simplest procapsid obtained is P14 from proteins P1 and P4, while P1 alone or with P7 yields unstable particles in cells that are fragile to isolation [19]. P1 can form a pentamer in the presence of P7, although P7 was not detected in crystals [37]. The bacteriophage Φ8 P1 structure was solved as a pentamer without any other proteins [38]. A minimal first product is likely a pentamer of P1_A_ with P4. The C-terminal tail of P4 extends beyond the P1_A_ to P1_B_ [48], suggesting that it could stabilize the addition of P1_B_ subunits to the pentamer.

Both P2 and P7 accelerate procapsid assembly [59,60]. P2 incorporates in a dose-dependent manner [59], consistent with the random placement at the 20 binding sites [33]. If P2 and P7 are not in the mixture before assembly, they are excluded if added afterward, consistent with their internal location [43]. P2 binds between three invaginated five-fold vertices (Figure 4D) [47] while P7 covers an interface between P1_A_ subunits (Figure 5A) [35]. The presence of all three accessory proteins, therefore, aids in the efficient production of the P1 shell and acts as the scaffolding commonly observed in other viruses [64].

The sequence of assembly events is uncertain beyond the initial pentamer, likely with a P4 hexamer and 2–3 copies of P7. The potential next step is the addition of P1_B_ subunits with the involvement of the P4 C-terminal tail. Conceptually, the P1_10_P4P7_2–3_ units could then come together to form the icosahedral shell with P2 binding between some neighboring P1_A_ pentamers.

## 4. Packaging of ssRNA Segments and Capsid Expansion

The packaging of plus-strand RNA in the cystoviruses differ from their eucaryotic counterparts [3]. Cystoviruses package in typical bacteriophage style by insertion into a preformed procapsid driven by a motor protein. Instead, in the *Reoviridae*, the ssRNA first assembles into a particle serving as the nucleating core for the assembly of the capsid shell [65].

From the early cryoEM work on Φ6, it was clear that the procapsid expands to assume its mature form in the nucleocapsid [30]. The internal volume increases by ~2.5 fold to accommodate the full dsRNA genome [31,45]. Concurrent with expansion, the genomic ssRNA segments package in a strict order, with the s segment first, followed by m and l [20]. The specificity is dependent on conditions with higher magnesium (>4 mM) and phosphate (>1.5 mM) concentrations, rendering the packaging non-specific. Once the l segment is packaged, the polymerase replicates the segments to produce the dsRNA genome [9].

The sequential order of packaging indicates that there should be two expansion intermediates, but only one was identified in the original structural study [30]. In a study where nucleocapsids lost their dsRNA, two partially contracted P1 shells were reconstructed [45]. These were interpreted as lower energy forms of the P1 shell, with the more contracted one indistinguishable from the expanded procapsid and denoted intermediate 1. The intermediate 2 map was constructed from a significant number of particle images indicating it as a clear state. Intermediate 1 is readily obtained by exposing the procapsid to mild heat, acid, or ionic strength conditions [45,46], indicating that the first expansion energy barrier is relatively low (Figure 7). The presence of a 10 × excess of P2 in assembled procapsids appears to prevent expansion [59]. This suggests that the binding of P2 between the invaginated five-fold vertices is strong enough to stabilize the procapsid. On expansion to intermediate 1, P2 is not apparently associated with the P1 shell anymore [45]. The ease with which intermediate 1 is obtained means that it is the more stable conformation of the P1 shell (Figure 7). Presumably, the packaging of the s segment is enough to trigger the first expansion event. The low efficiency of packaging [66] impeded efforts to observe the entry of the s segment by cryoEM.

The current understanding is that the procapsid exists in a metastable state that is easily triggered to expand to intermediate 1, either by acid, heat, or salt, and potentially by the packaging of the s segment generating some pressure inside the procapsid (Figure 7) [45]. The packaging of the m and l segments then increases the pressure on the P1 shell, with it reaching the fully mature state and pressurized state after replication. In reconstructions of the nucleocapsid, the regions around the 5-fold vertices are still flexible, indicating that they do not reach the level of stability of the rest of the capsid.

The demonstration that only one P4 hexamer per procapsid is necessary for packaging followed by minus-strand synthesis (replication) led to the proposal that there is a special vertex different from the others [67]. One possible consequence of such a special vertex is that one would expect there to be a correlation between the locations of P4 hexamer on the outside and the polymerase on the inside. No such correlation was found in a tomographic study of procapsids where individual proteins were counted [33]. It was concluded that such a special vertex is unlikely.

## 5. Completing the Virion

### 5.1. dsRNA Packaging of the Nucleocapsid

Remarkably, the dsRNA genome packages in a few configurations amenable to reconstruction such that the helical strands are clear (Figure 8A,B) [68]. In the cypoviruses, defined conformations of the dsRNA have also been found [69,70]. This means that the icosahedral capsids of the dsRNA viruses form very specific interactions with the genome. This is despite the current understanding that the packaging mechanisms are different. While the cystoviruses package their three ssRNA segments into preformed procapsids, the eucaryotic dsRNA viruses form their capsids around preassembled particles containing the 9–12 ssRNA segments within organized cellular structures called viral factories or viroplasms [3]. How these particles come together is complex and still requires intensive study [71].

While the dsRNA in Φ6 is arranged in ordered shells (Figure 8B), P2 and P7 are apparently not in detectable locations [35,47]. This contrasts with the cypoviruses where 10 transcriptional enzyme complexes, each composed of a polymerase and a packaging NTPase, locate to specific sites [69,70]. Somehow, during Φ6 transcription inside the nucleocapsid, P2 should be positioned at a five-fold vertex to allow a newly synthesized ssRNA transcript to be ejected by P4 on the outside. In bacteriophage Φ12, P4 acts as a passive conduit for the new transcripts [72]. Also, in Φ12, a density that could be P2 was shown underneath the five-fold vertices in virions [62]. Since there are only three segments, conceptually requiring up to three polymerases, they may not be distributed in regular locations conducive to the extensive averaging in cryoEM reconstruction.

### 5.2. The Intermediate Layer Composed of P4 and P8

The P8 layer is composed of 200 trimers arranged in a T = 13 *laevo* quasi-symmetric lattice with openings at the five-fold vertices to accommodate the P4 hexamers (Figure 8C,D) [48]. The interactions between the trimers show complicated domain-swapping features.

The P4 occupancy in the nucleocapsid is close to stoichiometric [43,68], indicating that it is assembled as such in the procapsid. The observation of lower amounts on purified procapsids should, therefore, be considered an artifact of purification. The Φ8 P4 hexamer is offset by 8 Å from the five-fold axis of the P1 shell [44], likely the result of the symmetry mismatch with only up to five of the six C-terminal tails binding it to the P1 shell [48].

The assembly of the 16 kDa P8 protein onto the nucleocapsid requires calcium (II) Ca^2+^, and it can be stripped off by removing Ca^2+^ [14]. Purified P8 can also self-assemble into spherical shells in the presence of Ca^2+^, but mostly without closing [14,73]. The location where Ca^2+^ binds is not known. The assembled nucleocapsid with P8 is infective in cells where the outer membrane and peptidoglycan layers have been disrupted [14]. P8 binds to host phospholipids [74], indicating a mechanism for enveloping newly formed nucleocapsids. During infection, P8 interacts with the cell membrane to form invaginations around the nucleocapsid [75]. The nucleocapsid coated with P8 has significantly lower polymerase activity compared to the procapsid [14], such that its uncoating and degradation promote transcription [75].

### 5.3. The Envelope

The envelope of Φ6 is the least studied structurally because it does not conform to the icosahedral symmetry of the inner two layers. It is composed of host lipids [76] and the proteins P3, P5, P6, P9, and P10 [77,78] (Figure 1) and requires P12 for proper formation [21]. P5, the lysozyme of Φ6, is thought to be located between the envelope and the P4–P8 shell [79]. P9 is the major integral membrane protein, accompanied by P6 and P10 [79,80]. In *E. coli*, P9 expression leads to the formation of P9-containing vesicles, with P12 playing a protective role in preventing the proteolytic degradation of P9 [81]. The most prominent feature of the envelope is the mushroom-like spikes protruding from the surface, identified as P3 and anchored to the membrane by P6 [82]. The equivalent multichain P3 of Φ12 is a hexameric spike (Figure 1) [17].

## 6. Coopting the Host Factor YajQ for Transcription Regulation

After breaching the bacterial envelope of its host and losing the P8 layer, the Φ6 nucleocapsid transcribes its genome to produce the three ssRNA segments [83]. It was noticed that nucleocapsids produced in vitro also show suppressed expression of the L segment, unless manganese was included [84,85,86]. The l segment differs from the s and m segments in the second nucleotide: it is 5′-GU rather than 5′-GG. This renders the nucleocapsid unable to transcribe the l segment or any segment starting with anything other than 5′-GG [9,58]. Mn^2+^ at 1 mM overrules this blockage by increasing the permissivity of the polymerase [9,54,55]. Mn^2+^ competes with Mg^2+^ at the non-catalytic metal ion binding site in the polymerase [57]. However, the concentrations of Mn^2+^ in the host do not reach the levels required for activating the polymerase, indicating that it is not the factor stimulating l segment transcription.

Qiao et al. [15] discovered that the nucleocapsid binds an 18 kDa host protein, YajQ, required for full transcription of the L segment. A similar regulation mechanism for L transcription was described for another cystovirus, Φ2954, except that it requires a different host protein, GrxC [87]. YajQ homologues are ubiquitous and conserved in bacteria [88]. A YajQ-GFP fusion protein coated infecting Φ6 nucleocapsids to such an extent that the individual particles were visible inside cells [89]. Later in infection when many new nucleocapsids are produced, the supply of YajQ is insufficient, yielding fewer l transcripts. This is interpreted as a fortuitous regulatory function to divert resources to the other segments during final virion assembly. YajQ binds to the outside surface of the nucleocapsids and dissociates in higher salt concentration [15]. Because YajQ does not activate the polymerase directly [58], it must act through the capsid shell with the polymerase on the inside.

Purified nucleocapsids decorated with YajQ (Figure 9A) were visualized by cryoEM (Figure 9B) [90]. The micrographs show many filled capsids, but also empty P1 shells likely derived from packaged capsids that lost their RNA content (Figure 9B) [45]. YajQ binds as a monomer to 60 sites close to the three-fold axes of the icosahedral P1 shell, in this case with an occupancy of ~58% (35 out of 60 sites) (Figure 9C). A homology model of YajQ based on the crystal structure from *Haemophilus influenzae* [91] fits unambiguously into the 3D reconstruction, indicating conservation of the fold (Figure 9D). The YajQ structure features two topologically similar domains connected by two linkers. Each domain has a beta sheet on one side and two alpha helices on the other. Each YajQ molecule straddles the two P1 subunits in the asymmetric unit, covering part of the C-terminal tail of P4 (purple density in Figure 9D).

Purified polymerase still shows discrimination against the l segment [24], but YajQ has no effect on it [58]. The evidence points to the stability of the C-terminal domain and its role in the self-priming initiation of transcription. Manganese relieves the requirement of the Φ6 nucleocapsids for YajQ, decreasing template specificity [58]. Substitutions in two of the genes confer YajQ independence: I632V in P1, and K34N and I642V in P2 [15]. Other substitutions in the hinge region of the C-terminal domain (V603A, A604E, and L612R) increase activity in nucleocapsids in the absence of YajQ, presumably by destabilizing this domain [58].

An additional density was noted close to where YajQ binds (Figure 8D) [90] that turns out to be the C-terminal tail of P4 first described by Sun et al. [48]. This may suggest a way of action. YajQ may stabilize the tail of P4 to such an extent that it affects the expression of the l segment. If so, the slight change at the 5′ end of the l transcript may pose a higher barrier for transcription.

## 7. Broader Impact of the Studies on the Structure and Transformations of Bacteriophage Φ6

We have made significant progress in understanding the Φ6 life cycle, but much remains to be discovered. This relatively simple system provided some answers about dsRNA viruses, and the structural studies benefitted from recent technological advances, contributing to the development of new experimental approaches.

An important question is how a virus with a segmented genome ensures that a full complement is present in the assembled virion. Many RNA viruses with segmented genomes can package them effectively to ensure all segments are accounted for to produce an infectious particle [3,92]. In Φ6, this is achieved by the controlled sequential packaging of the three segments [93], resulting in highly structured genome packing (Figure 8A,B). This is reminiscent of the packaging in dsDNA bacteriophages into a preformed head [94]. In contrast, the current model for the packaging of eukaryotic dsRNA viruses is a coalescence of the segments into a nucleoprotein particle that is then wrapped into a capsid shell in the reoviruses [3,92] or encased in a lipid envelope in the influenza virus [95]. In all cases, packaging signals in the form of RNA structural motifs direct the proper packaging process [93,96].

Many icosahedral virus capsids can self-assemble, often with the help of scaffolding partners [97,98]. In Φ6, the scaffolding role for P1 assembly is assumed mainly by P4, with assistance from P2 and P7 [19,59,60]. As with many procapsids, it assembles in a metastable state that can transform into a more stable mature form in Φ6, concurrent with packaging and expansion (Figure 7).

The studies of the Φ6 procapsid and nucleocapsid structures grew with the development of better capabilities in electron microscopes as well as computational approaches to processing micrographs. The discovery of the locations of the P2 and P7 proteins was directly related to better reconstruction of cryoEM maps and examining difference maps between procapsids with different compositions [32,34,35]. Knowing these locations, we could count the presence of the molecules using cryo-electron tomography and obtain detailed distributions of occupancy [33]. New techniques to handle symmetry mismatch were developed for understanding the structures of P4, P2, and dsRNA within the context of the procapsid and nucleocapsid [47,48,68]. We combined X-ray crystallography of the P1 pentamer with the high-resolution cryoEM maps of the procapsid and nucleocapsid to visualize the conformational differences between subunits and the transformations during capsid expansion [37]. We now have tools to study viruses in great structural detail, so the success of a project is largely determined by the quality of sample preparation.

The remarkable progress in structural biology has given us a detailed understanding of the dsRNA and two inner layers of the Φ6 virion and its transformations during its life cycle. Nevertheless, there are many questions that remain to be answered. The full sequence of events in procapsid assembly would contribute to resolving how complete icosahedral particles form. How do the three ssRNA segments recognize the conformational state of the procapsid? How is the nucleocapsid enveloped in the final stages of virion formation? The detailed structures of the integral proteins in the envelope and the spike still need to be determined. What triggers the lysis of the host cell? Structural details of the infection events are required to better understand how it binds to pili and breaches the cell envelope. After infection, is the P4 hexamer involved in the transmission of control from YajQ to the polymerase inside the P1 shell? There is ample room for further structural studies of the cystoviruses for their own sake but also in the broader context of protein interactions and dynamics, and not only pertinent to viruses.

## Figures and Tables

**Figure 1 viruses-15-02404-f001:**
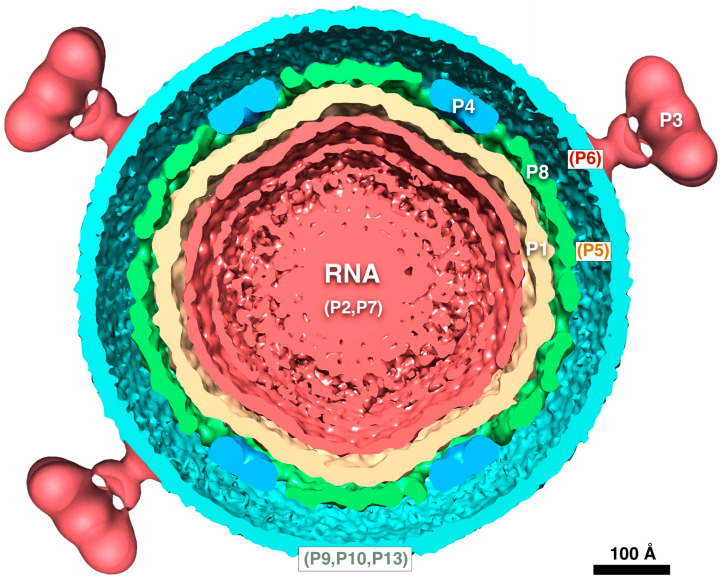
The bacteriophage Φ6 virion. The dsRNA is packaged within an icosahedral P1 shell together with proteins P2 and P7. Proteins P4 and P8 form an intermediate layer, surrounded by the envelope featuring the spike protein P3. Proteins in brackets have only approximate locations within the virion. (Map based on EMDB 1301 [16] and the spike from Φ12 [17]).

**Figure 2 viruses-15-02404-f002:**
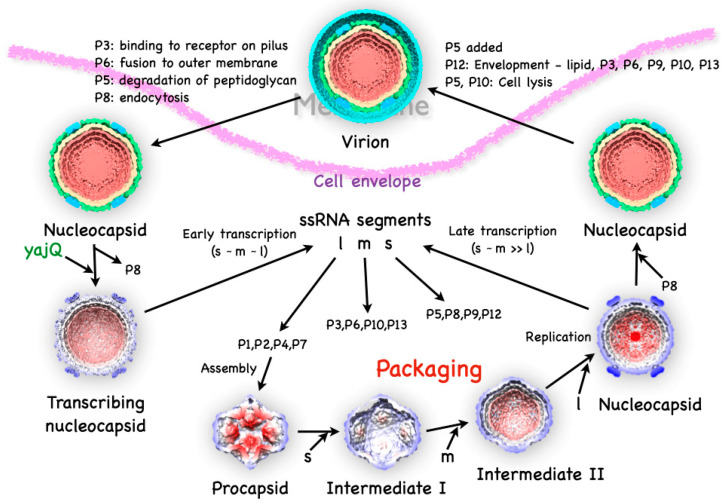
The bacteriophage Φ6 life cycle in structures. The virion has three layers surrounding the tripartite dsRNA genome (red): the capsid (beige), the intermediate layer (blue and green), and the proteolipid envelope (cyan). Infection starts when the P3 spike binds to the pilus receptor. The envelope then fuses with the outer membrane, degrades the peptidoglycan layer (P5), and enters, mediated by P8. After P8 disassembles, the host factor YajQ binds to facilitate the transcription of the l segment, so that it is expressed at about the same levels as the s and m segments. The l segment codes for the four proteins assembled into the initial product, the procapsid. The ssRNA segments are then packaged in sequence with corresponding expansion of the capsid shell. Once all the segments are packaged, replication completes the production of the dsRNA genome. The P8 shell is assembled, the envelope added, and the new virion released through cell lysis.

**Figure 4 viruses-15-02404-f004:**
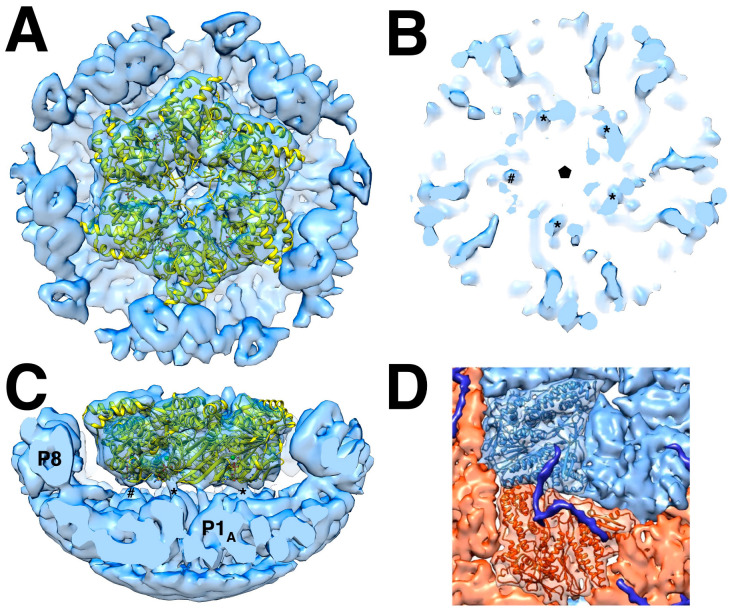
The packaging motor, P4: (**A**) The P4 X-ray crystal structure (PDB: 5MUV) fitted into the density of an asymmetric reconstruction of the five-fold vertex of the nucleocapsid (EMDB: 3573), surrounded by the P8 layer. (**B**) A section through the reconstruction at the contact plane reveals four possible connections (*), with the fifth (#) not quite forming a connection. (**C**) Side view showing the connections between P4 and the capsid shell below. (**D**) The C-terminal tail of P4 (blue) traverses the surfaces of P1_A_ (light blue) and P1_B_ (red) [48].

**Figure 5 viruses-15-02404-f005:**
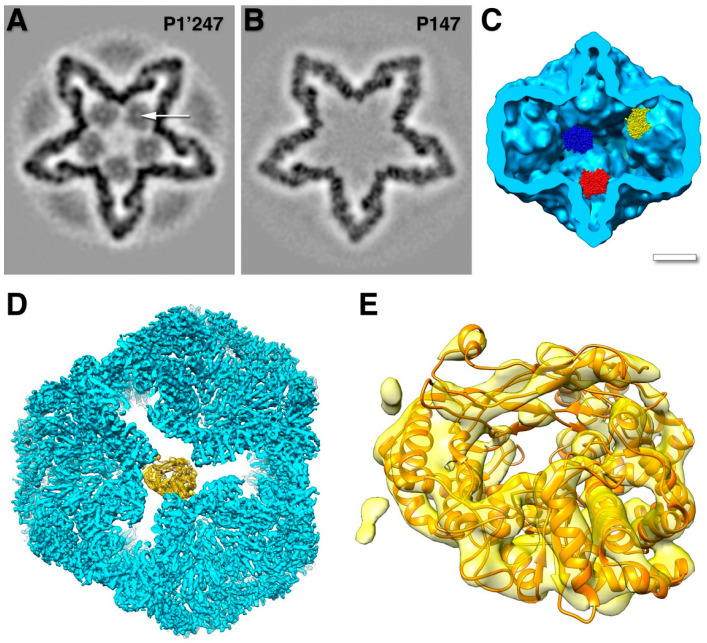
The location of the polymerase in the procapsid. (**A**) Section through a reconstruction of the procapsid showing the location of the polymerase (arrow). (EMDB: 1501) (**B**) Section through a reconstruction of a procapsid lacking the polymerase. (EMDB: 1502) (**C**) Isosurface rendering of a cut through the P1 shell of the procapsid with three polymerase atomic models arranged in the locations indicated in (**A**) [32]. Scale bar: 100 Å. (**D**) The polymerase (yellow) within the P1 shell (blue) viewed down a three-fold axis. (EMDB: 3185) (**E**) Fit of the polymerase crystal structure within the reconstructed density within the context of the P1 shell (EMDB: 3186; PDB: 5FJ6) [47].

**Figure 6 viruses-15-02404-f006:**
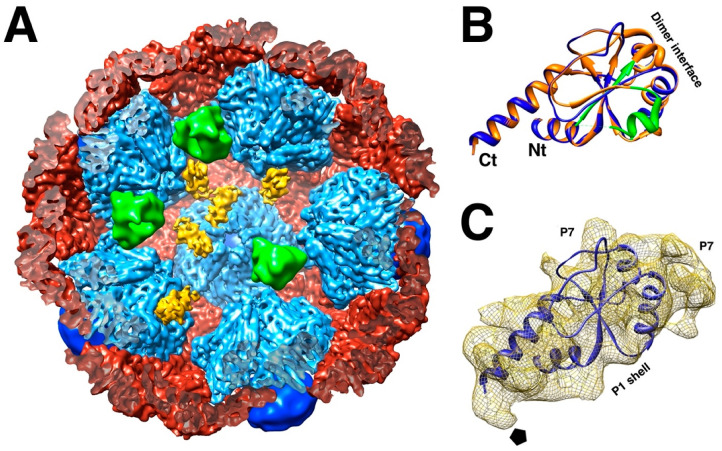
The location of P7. (**A**) Each P7 monomer (yellow) is bound to the interface between two P1_A_ subunits (light blue), but only at some locations around the five-fold axis. Polymerase monomers (green) are located at other sites that overlap with the P7 sites. The P1_B_ subunits (red) and P4 hexamers (dark blue) complete the procapsid. (EMDB: 2341) (**B**) Homology model of the Φ6 P7 based on the crystal structure of the Φ12 P7 protein N-terminal fragment. The dimer interface in the crystal is indicated. (**C**) One P7 density with a fit of the homology model. Also indicated are the interface to the P1 shell and the interfaces to two potential P7 neighbors around the three-fold axis. The pentagon denotes the five-fold axis [35].

**Figure 7 viruses-15-02404-f007:**
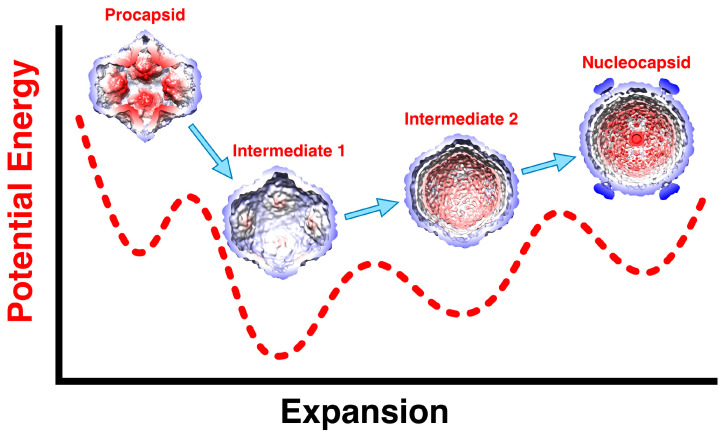
The transformations of the P1 shell on maturation with the dashed curve showing a conceptional energy profile along the expansion coordinate. The procapsid readily converts to the first expansion intermediate by various triggering events such as heat, acid, salt, or the packaging of the s segment. Packaging the m segment then leads to further slight expansion to the second intermediate, followed by packaging the l segment. Only then is the genome replicated to produce the full dsRNA complement.

**Figure 8 viruses-15-02404-f008:**
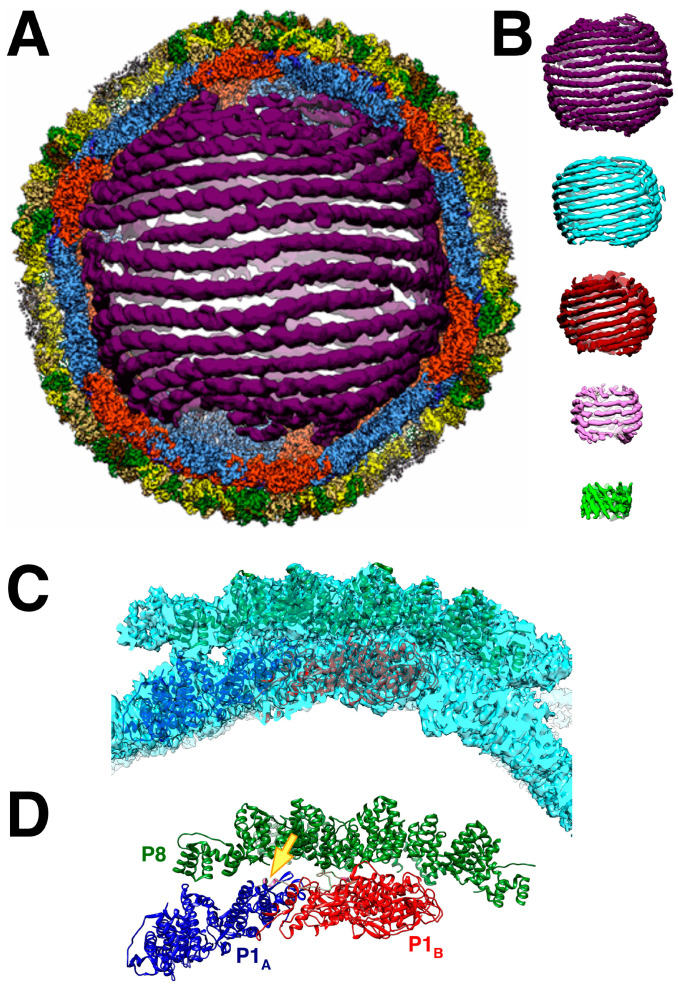
(**A**) The nucleocapsid orientated down a two-fold axis, showing a cut through the outer two layers and the first shell of dsRNA. The outer shell is the P4-P8 layer, but in this cut view, only the P8 layer is visible (green). The capsid shell is composed of the P1_A_ (blue) and P1_B_ (red) proteins. (**B**) The five dsRNA shells. (**C**) Two P1 subunits and 10 P8 monomers fitted into a reconstruction of the nucleocapsid. (**D**) As in C but without the density. The arrow points to the P4 C-terminal tail (pink) bound to the two P1 subunits [68].

**Figure 9 viruses-15-02404-f009:**
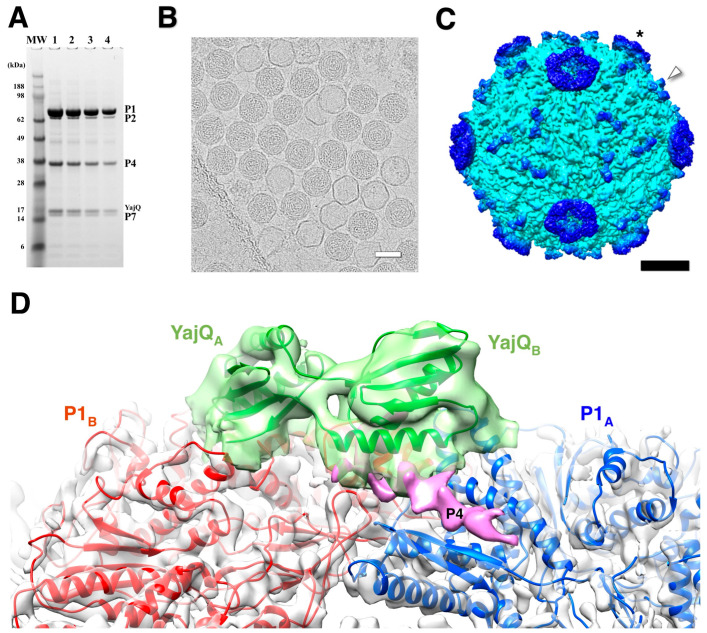
The bacteriophage Φ6 nucleocapsid with bound YajQ. (**A**) SDS-PAGE gel showing the four constituent proteins together with YajQ at a slightly larger amount than P7. (**B**) A representative micrograph of YajQ-bound packaged capsids with a few empty P1 shells. Scale bar: 500 Å. (**C**) A reconstruction of the nucleocapsid filtered to 6 Å. The YajQ monomers (arrowhead) are bound to sites surrounding the three-fold vertices. The P4 hexamers (asterisk) are the donut densities suspended above the five-fold vertices. Scale bar: 100 Å. (**D**) Fit of a homology model of YajQ (green) into the density attached to the capsid. Also shown are the two capsid subunits, P1_A_ (blue) and P1_B_ (red), and a purple density consistent with the C-terminal tail of P4 [90].

## Data Availability

All structural data used in preparing the review can be found in the Electron Microscopy Databank (https://www.ebi.ac.uk/emdb, accessed on 23 October 2023) and the Protein Data Bank (https://www.rcsb.org, accessed on 23 October 2023).

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
