# Peer review of "Structural Studies of Bacteriophage Φ6 and Its Transformations during Its Life Cycle"

_viruses, 2023, doi:10.3390/v15122404_

Round 1

Reviewer 1 Report

Comments and Suggestions for Authors

Comments on the Quality of English Language

Reviewer 2 Report

Comments and Suggestions for Authors

The manuscript “Structural studies on the bacteriophage Φ6 and its transformations during its life cycle” is a review of the 50th anniversary of the phi6 discovery. The author has conducted a very thorough and comprehensive review of the phi6 structural studies. The review can be very useful for the further researches of phi6, as well as its application in biotechnology.

I just have some minor comments:

- In the MS text, the author uses both names of RNA segments written with a lowercase letter (examples: lines 59-62, figure 1, line 375, caption to figure 6, ect) and with a capital letter (example: lines 48-49, line 372, ect). I suppose it is more common to write the names of RNA segments with a capital letter: S, M, L.

-Lines 165-175, lines 373-378, line 405. It would be more correct to replace the word “manganese” or symbol “Mn” with “manganese(II) ions” or with “Mn2+”. The same issue is with the Mg symbol (line 377).

- I didn't find any mention of Figures 7C and 7D in the MS text.

-It may be worth adding the names of the structures shown in the figure 6: Procapsid, Intermediate I, Intermediate II….

-Line 338, A space is needed between “13” and “laevo”.

-The names of viruses’ families should be written in italics.

Reviewer 3 Report

Comments and Suggestions for Authors

Peer Review for Viruses

Heymann

Review article

Title

Structural studies on the bacteriophage phi6 and its transformations during its life cycle.

Summary

Bacteriophage phi6 is an attractive model system that includes a viral envelope, dramatic conformational shifts, and discrete functional regulators. This manuscript reviews the structural transformations of phi6, as it progresses through its lifecycle, and the physical or chemical communication of checkpoints that enable the virus to respond to each stage. As with all great reviews, it raises new questions at the edge of the known field.

Major comments

This manuscript is delightfully well-written. I have no major comments.

Minor comments

Page 2 and 12: Segment capitalization is variable, mostly lowercase but at least two instances of uppercase. Typically not a big deal but in this font the lowercase looks like the number 1, and the L segment figures prominently in the description of regulation and dosing.

Page 4: unformatted reference.

Page 6: spaces between phi 8 and 13.

Page 10: Eject or inject? Author’s choice here, but genome ejection is usually from the capsid and here it is unclear if this is ejection from the replication complex but into the capsid during packaging. I am not following the direction of operations in this passage, please clarify.
